# Analysis of the Impact of Selected Parameters of the Hybrid Extinguishing System on the Fire Environment in a Closed Room

**Jerzy Gałaj** [1,*], **Tomasz Drzymała** [2] **and Paweł Wolny** [3]

1  Institute of Safety Engineering, The Main School of Fire Service, Słowackiego Str. 52/54, 01-629 Warsaw, Poland

2  Faculty of Safety Engineering and Civil Protection, The Main School of Fire Service, Słowackiego Str. 52/54, 01-629 Warsaw, Poland; tdrzymala@sgsp.edu.pl

3  Independent Research, 05-091 Ząbki, Poland; p.wolny@wp.pl

*  Correspondence: jgalaj@sgsp.edu.pl; Tel.: +48-693-175-252

**Abstract:** The main purpose of this study was to analyze the impact of some parameters (water mist flow rate and type of gas used) of the hybrid extinguishing system on the fire environment (temperature as well as carbon monoxide and oxygen concentrations) in a closed room. Hybrid fire-extinguishing systems in which water mist is driven by inert gas combine the advantages of typical fog systems and fixed gas extinguishing devices. They have been developed in the last years but are now being used more and more often and the preparation of standards for them is planned for 2020. For this purpose, many fire tests with this system should be conducted. Some of them are discussed in this paper. Two different flow rates of water mist (1.5 or 3 dm$^3$/min) and inert gas (nitrogen or air) were used during hybrid system testing. Some parameters of the fire environment in the compartment such as temperature measured by thermocouples as well as carbon monoxide and oxygen concentrations measured by electrochemical gas sensors are presented here. The characteristic values of the extinguishing process are also included. The assumed times of ensuring safe conditions in the room have been confirmed.

**Keywords:** hybrid extinguishing system; internal fire; fire extinguishing; water mist; fire environment; carbon monoxide concentration; oxygen concentration

## 1. Introduction

Hybrid fire-extinguishing systems combine inert gas and water mist. This innovative technology is currently being implemented and used in fixed fire-extinguishing devices. The term fixed extinguishing devices means fire-fighting devices permanently connected to a building structure, activated automatically in the early phase of the fire, having their own extinguishing agent supply and transmitting information about the fire to a dedicated place. The literature available in source materials around the world describes the issue related to hybrid fire-extinguishing systems in a fragmentary way. Studies are currently underway on a separate NFPA (National Fire Protection Association) standard for this type of solution in the USA. According to the provisions of the draft standard being developed [1], a hybrid fire-extinguishing system is a fire-extinguishing system capable of supplying water and inert gases with a given expense and proportion. This mixture is assumed to be the most effective combination of both components, where both extinguishing agents take part in the extinguishing process. Works on the development and implementation of the NFPA 770 standard are to be completed in April 2020. According to the concept adopted by the members of the NFPA Technical Committee, hybrid fog differs from a twin fluid water mist system in that both gas and water are extinguished in

the hybrid system, and in the second case, water is the only extinguishing agent. The gas serves only as a medium for spraying water [2].

Interest in implementation on an industrial scale of systems combining water mist and gaseous extinguishing agents appeared in 1996. It was found that a combination of water mist extinguishing using inert gas for propulsion could increase extinguishing efficiency. In 1996, Zhigang Liu and Andrew K. Kim, while researching water fog, tested a hybrid system that was classified as "Water fog with additional medium" [3]. The results of the experiments confirmed that the mixture of water mist and inert gases improves the extinguishing efficiency compared to the water mist alone. It was also found that when using a hybrid system, it can prevent the re-ignition of combustible materials, while the presence of inert gas can reduce the emission of acidic fire gases. It has been mentioned that the extinguishing efficiency is strictly dependent on the correct placement of the nozzles. In addition to the above studies, the amount of available literature on hybrid water mist systems up to recent years was negligible. The popularity of this solution has increased significantly in recent years due to its wide spectrum of use and functionality. Systems from two companies appeared on the American market—Victaulic and ANSUL (ANSUL is a global premium brand of Tyco Fire Protection Products)—which implemented hybrid fire-extinguishing systems based on a mixture of water mist and nitrogen [4]. In Europe, such solutions are presented by Siemens (Sinorix $H_2O$ Gas system) and the Italian company Tema Sistemi (Aquatech$^®$ Water Mist System). However, the last solution mentioned, according to the only currently functioning certification, as well as the initial findings of the NFPA standardization committee developing the standard and guidelines for fire tests, is not a hybrid solution [1].

Although there are studies in which attempts have been made to describe the theoretical problems associated with extinguishing fires in the post-flashover phase by means of spray jets, so far, very few experiments have been carried out to investigate their extinguishing effectiveness in various stages of internal fire development. One of the few examples is the result of tests for extinguishing fires with the help of jets from a high-pressure (20–30 bar) and low-pressure (6–8 bar) nozzle, discussed in papers [5–10]. The main conclusion of these studies is that the stream fed at a higher pressure is more effective (causes a faster drop in temperature and stream of radiated heat) and at the same time requires a lower water flow rate to achieve the same extinguishing effect. In previous studies, the impact of various parameters on the efficiency of fog nozzles has not been analyzed, and only some aspects related to the efficiency of water feeding have been addressed. Based on the review of the available literature, it can be concluded that the proposed tests are unique on a global scale in terms of their versatility, the use of various types of extinguishing devices and the number of parameters that will be analyzed.

The extinguishing efficiency of fog streams generated by the selected hybrid nozzle during extinguishing type A fire in a closed room was assessed. This study consisted of recording the temperature and toxic gas concentrations at selected points of the room during extinguishing burning wood. The parameters were registered from the moment of fire initiation to complete extinguishing. The obtained temperature during extinguishing will allow the validation of extinguishing models developed at the Main School of Fire Service, in which the fog nozzle was used as extinguishing equipment [11–14]. The work on the study of hybrid nozzles is a continuation of earlier work on water-jet generators [15,16] with known spraying spectra.

The fire can be divided into three basic phases, which differ in the rate of change in fire parameters over time. Research into the development of fires in a closed room leads to the conclusion that can be described by a change in the average gas temperature over time. The first phase is the period of fire growth, referred to in literature as the pre-flash or pre-flashover phase. From the moment the fire starts, the burning material heats the surroundings, causing the fire to spread. The fire temperature is low, and the combustion zone is located near the fire. The second phase is a period of fully developed fire, called the post-flashover phase. In this phase, all combustible materials in the room burn with almost equal intensity, which means that the temperature almost does not change.

The rate of combustion in this phase of the fire depends mainly on the size of the ventilation holes, i.e., the air supply to the combustion zone and the amount of combustible material. Flames fill the entire room and the gas temperature reaches its maximum value. Between the first and second phase, there may be a sharp change in the value of fire parameters and a change in the nature of the fire from local (surface combustion) to surface/spatial combustion, in which all combustible materials burn simultaneously. This transition is very short and is referred to as flashover [17–20]. In typical residential rooms, flashover occurs depending on their volume and flammable load during 10–20 min. Currently, for design purposes, depending on the type of combustible materials and fire scenarios, various temperature curves (including standard, external, hydrocarbon) are used—the description of which is given in the standard [21]. In the third phase of the fire, extinguishing occurs, characterized by a gradual decrease in temperature and smoke. Its beginning is assumed when the temperature reaches 80% of its maximum value. Thus, the fire is treated as all phenomena accompanying the combustion process. The basic fire parameters are quantities that are functions of time and space. The most important of them include the duration of the fire, fire surface, temperature, heat release rate, fire spread rate, mass burning rate and visibility range. During the fire, there are many different factors that threaten people's lives and health. Among them, one should first of all distinguish, heat emitting and accumulating in rooms during a fire, limiting visibility in areas of smoke accumulation, reducing oxygen concentration and toxic effects of secreted combustion products such as carbon monoxide, hydrogen sulfide, hydrogen cyanide, hydrogen chloride, nitrogen and sulfur oxides [22].

Firefighting using a hybrid system works on most of the above factors because it combines both the advantages of extinguishing with the help of water fog (the main purpose of which is to reduce the temperature in the combustion zone by receiving a large amount of heat) and extinguishing with the help of gas (the main purpose of which is to reduce the oxygen concentration and eliminate free radicals). In addition, gas in this case acts as a factor driving the water jet. Most materials in the fire zone, including wood, can contain chlorine from 0.01% to 0.8%, or even 1.75%. In the event of a fire impact on materials that constitute the equipment of housing and finishing materials in buildings, there is an emission of substances such as chlorine, hydrogen chloride or dioxins, cited in article [23]. Results of tests for dioxin emissions from house and office fires indicate a concentration level in fire smoke of several ng TEQ/$m^3$ (measure of dioxin and dl-PCB concentration expressed in so-called Toxic Equivalent). Combustion processes in the conditions of fire in the developed phase, at temperatures of 800–1000 °C, with insufficient oxygen, lead to the formation of incomplete combustion products—soot particles containing polycyclic aromatic hydrocarbons that react when cooled with water to a temperature of 400–600 °C, and react with chlorine. Under these conditions, cyclization processes with chlorine atoms take place, which leads to the formation of dioxins. The cooling effect leads to the recombination of a large number of radicals produced at high temperatures. These radicals have a major contribution to the formation of dioxins. To sum up, the synthesis of dioxins occurs through the reaction of chlorine with soot particles in the presence of oxygen and water vapor. Copper and aluminum oxides are catalytic. Rapid water cooling of the fire environment, from a temperature of 800 to 100–200 °C, causes the "freezing" of radical reactions, which reduces dioxin emissions. The method of feeding water through fog and hybrid systems is a great advantage in reducing dioxin emissions by rapid cooling of the fire environment, which is not available in other solutions used in fixed fire-extinguishing devices.

## 2. Materials and Methods

The subject of this study was a hybrid fire-extinguishing system, in which four two-phase atomizing heads of type FEN T were used, which can be simultaneously fed with water and inert gas. The technical data of the head FEN T are given in Table 1.

**Table 1.** Technical data of the head FEN T.

| Type of Technical Data | Gas | Water |
|---|---|---|
| Input pressure | 4 ± 0.5 bar | 4 ± 0.5 bar |
| Nominal flow rate | 0.5 ± 0.1 m³/min | 1 ± 0.1 dm³/min |
| Flow factor | 0.25 dm³/(min bar$^{0.5}$) | 0.5 dm³/(min bar$^{0.5}$) |
| Media cleanliness | solid particles d <40 μm particle density <10 mg/m³ | filter 300 μm |
| Power connections | External thread 1/4″ | External thread 1/8″ |
| Effective range of the stream | | 2.5 m |
| Maximum range of the stream | | 3.5 m |
| Droplets size | | from 4 to 200 μm |
| Operating temperature | | from 10 to 700 °C |
| Head mass | 0.4 kg | |
| Mounting | threaded hole M6 | |

The nozzle arrangement was mounted on a bipartite supply manifold allowing a significant orientation of each stream. The view of one of the nozzles is shown in Figure 1.

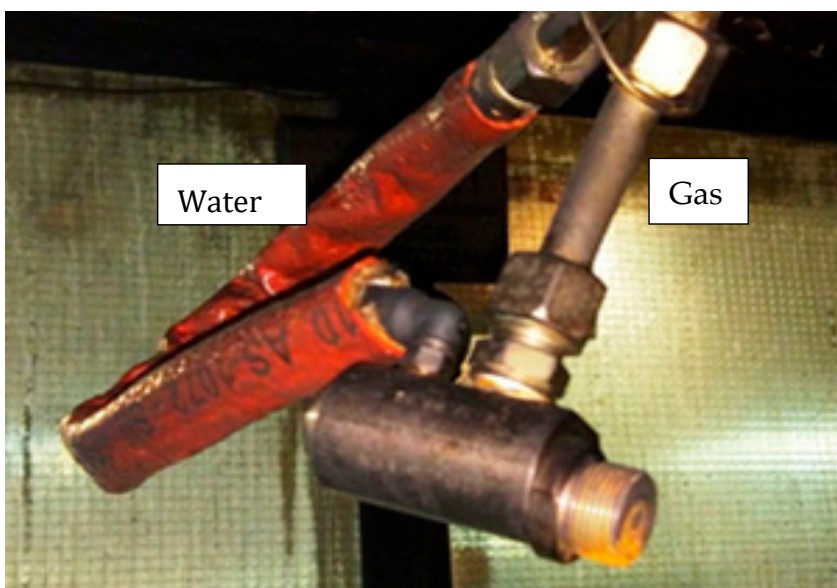

**Figure 1.** A view of a two-phase nozzle.

The tests were carried out in a closed room, with an area 25 m² (5 m long and 5 m wide). The height of the room was 2.8 m. Two walls were lined with ceramic tiles and two were made of tempered glass resistant to high temperatures. Water drainage was provided by a channel located in the center of the room. The water and gas installations were attached to the frame at a distance of 40 cm from the ceiling. The soft water pipes were secured with a fireproof sleeve. The water was fed using a pump set in an adjacent room. The gas installation was supplied from a bundle of cylinders connected in parallel, in which there was gas (air or nitrogen) at a pressure of 200 bar. It was fed through high-pressure pipes to the reducer, from which it was supplied to the fire-extinguishing nozzles under a pressure of 4 bar. Water and gas installations were made of steel pipes connected by a Parker system. It allows connecting rigid and flexible pipes. In order to achieve the maximum tightness of threaded connections, special sealing compound CX80 was used. A diagram of the fire-extinguishing system is shown in Figure 2. The blue line indicates the lines through which water is supplied and the green line through which gas is supplied.

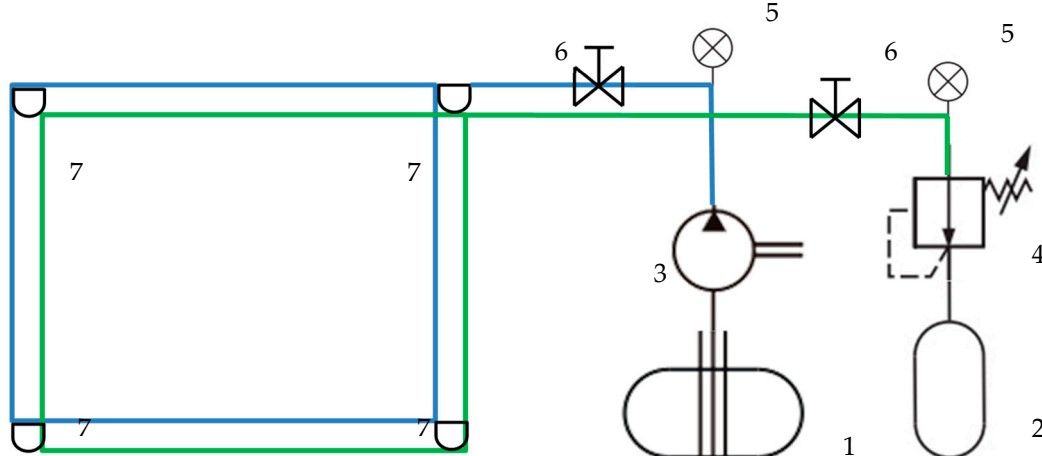

**Figure 2.** A diagram of the fire-extinguishing system (1—water tank, 2—gas tanks, 3—water pump, 4—gas pressure reducer, 5—manometers, 6—shut-off valves, and 7—nozzles).

The test room has equipped with a network consisting of 26 thermocouples–TP 212, with a length of 2.5 cm and a diameter of 1 mm. The thermocouple measuring range was from −20 to 1000 °C and its measurement accuracy was 1.5 °C in the range −20–370 °C or 0.4% in the range 371–1000 °C. All thermocouples are connected to a computer via Advantech ADAM measuring modules. Each thermocouple number (1–26) corresponded to the number programmed in Advantech ADAMView. Program start and recording of temperature readings began when the wood pile was set on fire. The measurement was finished when the flame of the burning material was suppressed. In total, thermocouples included in the measuring grid have been arranged so as to obtain reliable temperature distribution in the corner of the room where the fire source was located (the most unfavorable fire location from the point of view of fire extinguishing). Arrangements of thermocouples on two walls adjacent to the fire source (left and front) and on the ceiling are shown in Figures 3 and 4. All dimensions are given in centimeters.

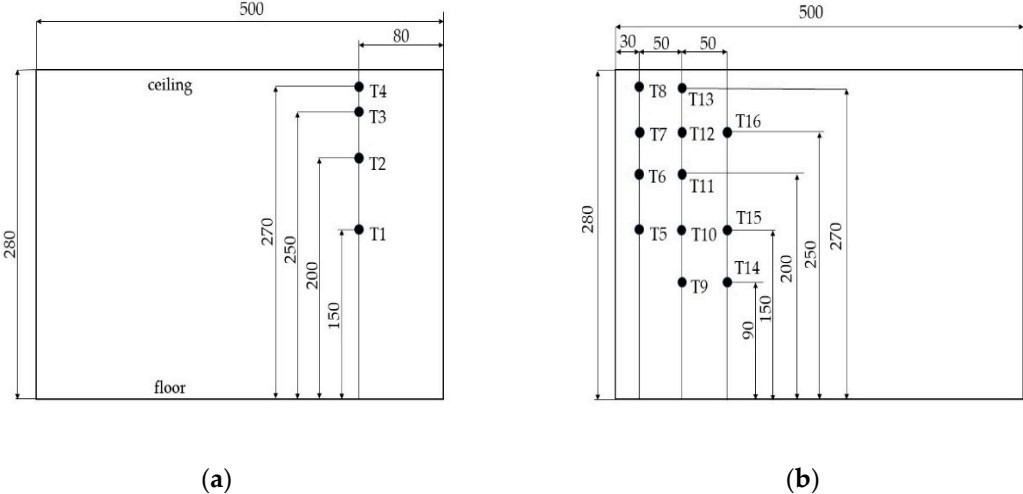

(**a**)  (**b**)

**Figure 3.** Arrangement of thermocouples on the wall located: (**a**) on the left side of the room; (**b**) in front of the room.

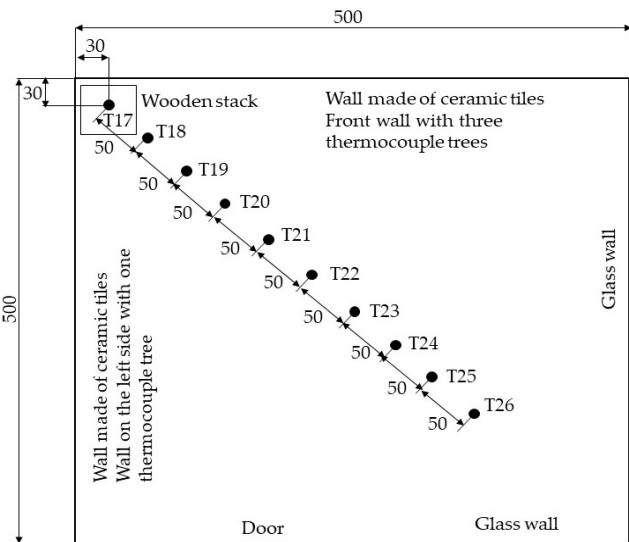

**Figure 4.** Arrangement of thermocouples on the ceiling.

Oxygen concentration sensors installed at two heights, 2.1 m (A1) and 1.7 m (A2), and one carbon monoxide sensor at a height of 2.1 m (B1) above the floor, were installed. Oxygen and carbon monoxide sensors were connected to the 16-channel measuring module, which transmitted the measured values to the computer. The MD-16 View 2.2 program read and recorded measurements every 2 seconds, saving each test in a separate file. Carbon monoxide measurements were made using a Gazex detector—model DG P2 E/N—with an electrochemical sensor with a range of 0–500 ppm, measuring with an accuracy of 1 ppm. Oxygen measurements were made using a Gazex detector— model DG P9 E/N. It was attached near the carbon monoxide sensor. The measuring range of this device is from 0% to 25%, with a resolution of 0.1%. The location of the carbon monoxide and oxygen sensors is shown in Figure 5, while their view is shown in Figure 6. Before each test fire, the efficiency of the installation and the readings from the measuring devices were checked.

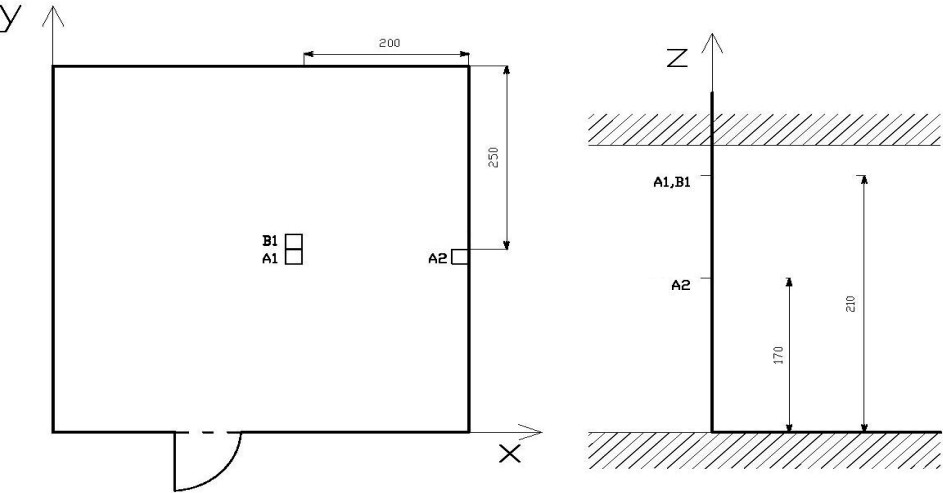

**Figure 5.** Horizontal (left) and vertical (right) sensor arrangement (A1, A2—oxygen sensors; B1—carbon monoxide sensor).

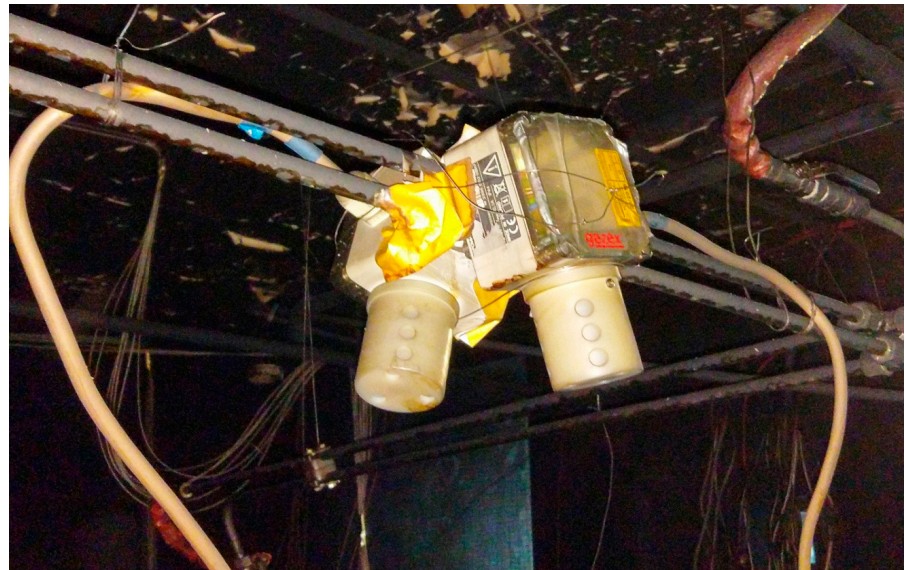

**Figure 6.** A view of oxygen and carbon monoxide sensors.

The main goal of the experiments was the determination of the time of effective extinguishing of flames and comparing extinguishing efficiency using a combination of water mist and gas in various proportions. Compressed air or nitrogen was used as the propellant gas. For each test, a pile of pine wood was created, consisting of 50 boards with a moisture content of 5%–15% and a density of approximately 0.55 kg/m$^3$, which was placed in the corner of the room. The staggered arrangement of the boards allowed free flow of air to the material being burned and the rapid development of flame combustion between the boards. The average stack weight was approximately 4.5 kg. This amount of pine wood allowed the generation of a test fire with an average power of approximately 100 kW, which was estimated on the basis of calorimetric tests of the combustible material used and its quantity The pile was placed at a height of approximately 40 cm above the ground. In order to achieve flame wood burning, approximately 250 ml of low boiling kerosene was poured into a metal tray on which the stack was placed. The fuel stacking scheme and two views of the stack—top and side—are shown in Figure 7.

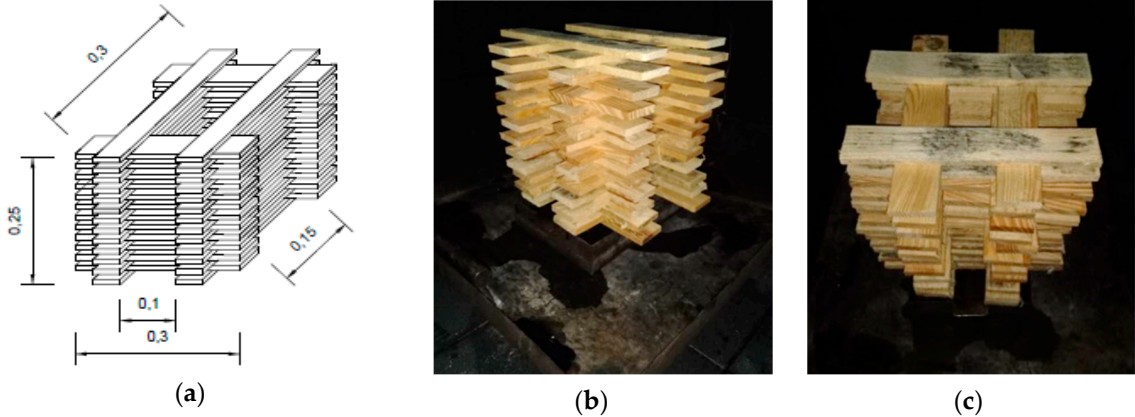

| **(a)** | **(b)** | **(c)** |

**Figure 7.** A wooden stack as a flammable material: (**a**) scheme of stacking; (**b**) side view of the stack; (**c**) top view of the stack [24].

After setting fire to the kerosene under the pile, the room door was closed. The room tightness was not measured in the work. However, even if there was a minimal air flow associated with door leaks and two relief openings (closed during the test), it could have had possible effects on the development

of the fire, but not on the extinguishing process to which this work relates. A recording of changes in temperature and gas concentrations were made on a computer stand. Extinguishing began when a temperature of 200 °C was reached on any thermocouple (when the stack was placed in the corner of the room, most often it was thermocouple T17, which was located directly above the fire at a height of 2.8 m). Extinguishing was terminated when the fire was completely suppressed and the temperature dropped below 150 °C to prevent the flame returning again spontaneously. The maximum extinguishing time was set at 10 min, taking as reference the extinguishing times of flammable liquids from the FM Global 5580 standard [25]. To ensure identical initial conditions in all tests, the room was ventilated and cooled to ambient temperature after each completed test. Extinguishing tests can be compared with each other, provided that the conditions for the development of the fire and the moment when extinguishing begins are identical. In trying to meet the first condition, testing took place in the same room with the same combustible material identically stacked and located in the same place. In trying to meet the second condition, extinguishing was started after reaching the same specific temperature at a given point of the room located in the ceiling above the combustion zone.

## 3. Results

The main parameter determining the extinguishing efficiency of the tested installation is the extinguishing time. A total of 37 tests were carried out on a proprietary fire-extinguishing system based on solutions used in two-media fog installations. Six of them were rejected due to a negative extinguishing result or excessive deviation of extinguishing start times from the average value for the remaining tests. First of all, they were to show that the gas used in the fog extinguishing system, in accordance with the NFPA 750 standard [26], is not only a driving factor, but also takes part in the extinguishing process. For this purpose, several tests were also carried out using water fog powered not only by nitrogen, but also by air. Given the lack of repeatability of the results caused, among others the complexity of the phenomena studied, and in many cases a small number of tests (in some cases the test was performed once) with the same hybrid system parameters (limited time and financial resources for research), it was practically impossible to carry out a full statistical analysis of the obtained results. Therefore, it was limited to providing the average value and standard deviation of the extinguishing time for various water capacities and inert gases used. For the same reason, only the tests where the water flow rate was 1.5 or 3 $dm^3$/min and the extinguishing time was closest to the average extinguishing time with the same parameters were selected for the analysis. In comparison, a test using pure nitrogen without water mist was also carried out. To systematize the obtained results, the following two points on the time axis and one time interval were distinguished:

- Start ($\tau_1$)—the time counted from the moment of ignition to the moment when extinguishing began, [s];
- Stop ($\tau_2$)—time counted from the moment of ignition to the moment when extinguishing was stopped, [s];
- Extinguishing time ($\tau_g$)—the difference between $\tau_2$ and $\tau_1$, [s].

The temperature of air in the selected point of the room measured by thermocouple No. 17 mounted below the ceiling at a height of 2.7 m, which was located above the water stream, was taken for analysis. Temperature curves obtained during the extinguishing test using water mist with different flow rates driven by nitrogen or air are shown in Figure 8. In order to analyze the temperature, the parameter to characterize its rate of dropping after extinguishing **v$_t$** was used, defined by the following expression:

$$v_t = (t_g - t_{min})/(\tau_{min} - \tau_1) \ [°C/s] \tag{1}$$

where:

$t_g$—temperature corresponding to time $\tau_1$ (start of extinguishing) [°C];
$t_{min}$—minimum temperature during drop phase [°C];

$\tau_{min}$—time corresponding to temperature $t_{min}$ [s].

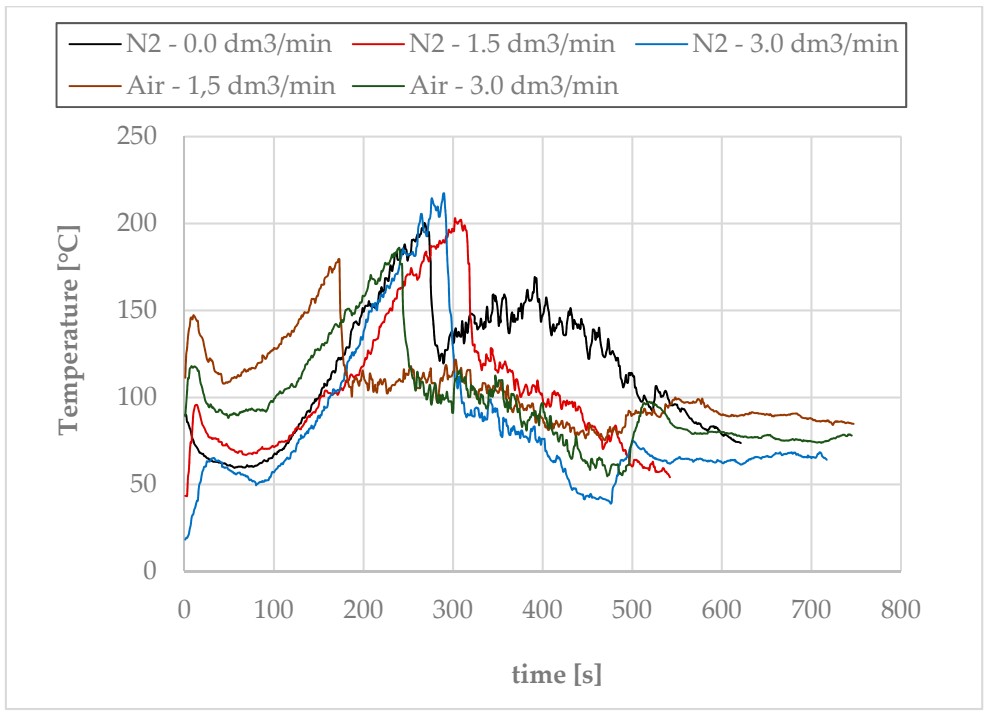

**Figure 8.** Temperature measured by thermocouple No. 17 during fire test extinguished with water mist at different flow rates driven by the inert gas nitrogen ($N_2$) or air.

From the point of view of the safety of people in the hazardous room and the development of fire, it is important to study the carbon monoxide and oxygen concentrations. The measurements of carbon monoxide and oxygen concentrations were not included in the standard assessment process of fixed fire-fighting devices. They were only intended to determine whether the change in volumetric water flow rate and type of propellant gas in hybrid systems have an impact on carbon monoxide emissions and oxygen consumption. For this purpose, carbon monoxide and oxygen concentrations are shown in Figure 9 (nitrogen) and Figure 10 (air). The times corresponding to the activation of extinguishing ($\tau_1$) were marked on the charts with vertical lines (the color of the lines is the same as the curves). Two critical levels of carbon monoxide concentration—100 and 500 ppm—were adopted, indicated by black horizontal dotted lines. The first of the results from the applicable legal acts regarding the level of the Maximum Permissible Instantaneous Concentration (NDSCh) contained in the Regulation of the Minister of Labor and Social Policy of 6 June 2014 on the highest allowable concentrations and intensities of factors harmful to health in the work environment (Journal of Laws of 2014, item 817) in the case of carbon monoxide is 117 mg/m$^3$, which is approximately 100 ppm after conversion. The second one corresponds to the maximum value used in tests in the MSDS (Material Safety Data Sheet) for this substance. At the same time, it is still considered safe for people staying briefly in a threatened room. The critical value of oxygen was a mass concentration of 15%—below which the combustion process practically stops due to too little oxygen and, moreover, it is dangerous to humans (marked on the charts with a green dotted horizontal line).

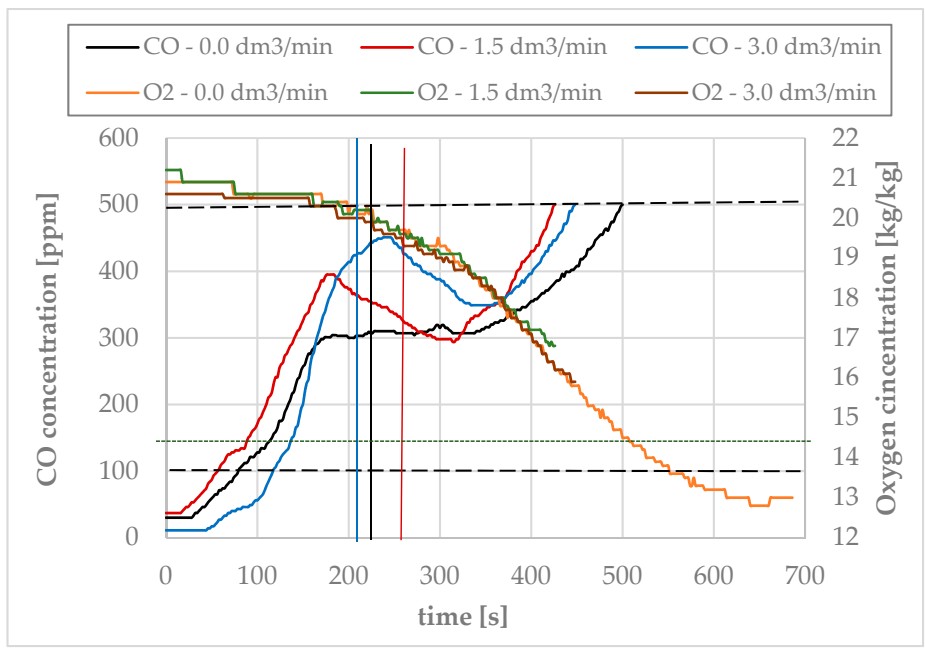

**Figure 9.** Carbon monoxide and oxygen concentrations measured during fire test extinguished with water mist at different flow rates driven by the nitrogen.

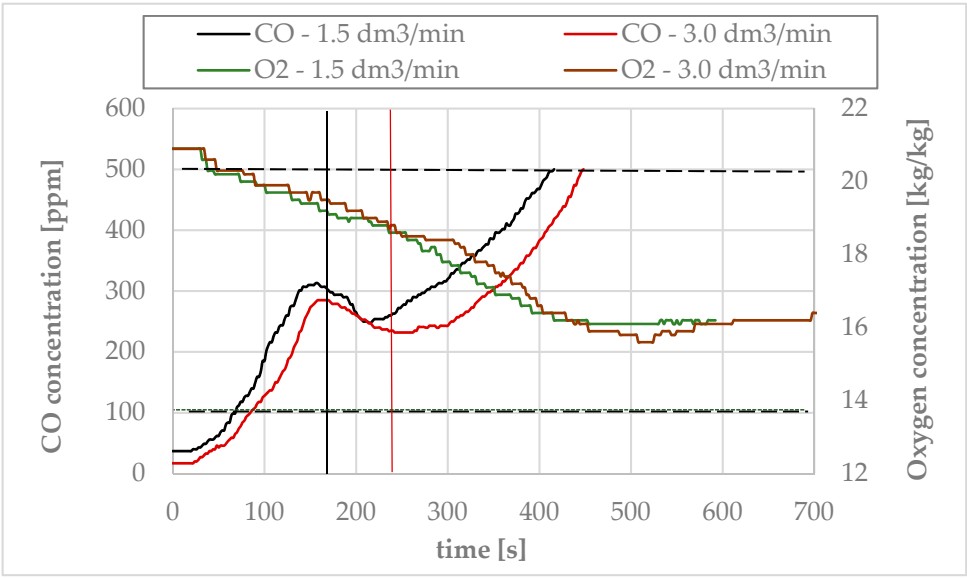

**Figure 10.** Carbon monoxide and oxygen concentrations measured during fire test extinguished with water mist at different flow rates driven by the air.

The average value $\overline{\tau_g}$ and standard deviation $\sigma_\tau$ of extinguishing times for individual parameters of the hybrid system are presented in Table 2. Here, the value of n means the number of tests carried out for the same gas and water flow rate.

**Table 2.** The average value and standard deviation of extinguishing times for different gas and water flow rates.

| No. | Gas | Water Flow Rate [dm³/min] | n | $\overline{\tau_g}$ [s] | $\sigma_\tau$ [s] |
|-----|-----|---------------------------|---|-------------------------|--------------------|
| 1 | $N_2$ | 0.0 | 1 | 349 | * |
| 2 | $N_2$ | 1.5 | 9 | 207 | 17.5 |
| 3 | $N_2$ | 3.0 | 8 | 198 | 21.8 |
| 4 | Air | 1.5 | 1 | 439 | * |
| 5 | Air | 3.0 | 2 | 242 | 18.0 |

* The number of tests is too small to determine the standard deviation.

## 4. Discussion

Based on the data included in Table 3, it can be stated that the longest extinguishing time equal to 439 seconds (over 7 minutes) was obtained by using air-driven water fog fed with a flow rate of 1.5 dm³/min, while the shortest time equal to 205 seconds (less than 4 minutes) was obtained with a flow rate of water fog equal to 3.0 dm³/min driven by nitrogen. A slightly larger extinguishing time of 223 seconds was also obtained using nitrogen, but with a lower water flow rate of 1.5 dm³/min. An approximately 55 seconds longer extinguishing time (260 seconds) was obtained when an air was used instead of nitrogen as the propellant gas at the same water flow rate equal to 3.0 dm³/min. During extinguishing with pure nitrogen, the extinguishing time was 349 seconds. It was 120 seconds longer than the extinguishing time with the use of water fog driven with this gas. Comparing the extinguishing times obtained, it can be stated that they are generally longer when using air as the propellant gas. This may result from a larger amount of oxygen in the space covered by the fire supplied with it.

**Table 3.** Selected output parameters of the temperature and gas concentration characteristics.

| No. | No. of Test | Gas | Water Flow Rate [dm³/min] | $\tau_1$ [s] | $\tau_2$ [s] | $\tau_g$ [s] | $v_t$ [°C/s] | $\tau_{CO\ 100}$ [s] | $\tau_{CO\ 500}$ [s] | $\tau_{O2}$ [s] |
|-----|-------------|-----|---------------------------|--------------|--------------|--------------|--------------|----------------------|----------------------|------------------|
| 1 | 20 | $N_2$ | 0.0 | 271 | 620 | 349 | 4.9 | 79 | 500 | 476 |
| 2 | 23 | $N_2$ | 1.5 | 303 | 526 | 223 | 5.8 | 57 | 426 | * |
| 3 | 1 | $N_2$ | 3.0 | 265 | 470 | 205 | 6.0 | 117 | 448 | * |
| 4 | 16 | Air | 1.5 | 169 | 608 | 439 | 5.0 | 69 | 416 | * |
| 5 | 15 | Air | 3.0 | 235 | 495 | 260 | 5.2 | 85 | 448 | * |

* Oxygen concentration did not decrease to 15% during the test.

Based on the temperature curves shown in Figure 8 and the data summarized in the Table 3, it can be concluded that the average rate of temperature drop is not directly related to the time of extinguishing. The average values of the temperature drop rates as a result of starting the extinguishing range from 4.9 to 6.0 °C/s. The slowest temperature decrease was observed during extinguishing with pure nitrogen and the fastest during extinguishing with water mist fed with a flow rate of 3 dm³/min, driven by nitrogen. The average temperature drop rates when extinguishing with air-driven water fog were approximately 0.8 °C/s lower than when extinguishing with nitrogen-driven water fog at the same flow rate (5.0 and 5.8 °C/s at 1.5 dm³/min as well as 5.2 and 6.0 °C/s at 3.0 dm³/min).

Based on the concentration curves shown in Figures 9 and 10, the changes in carbon monoxide concentration during the tests consist of several phases. In the first of them, there is a rapid increase in CO concentration to over 200 ppm, and then after reaching this value, the concentration decreases or in some cases remains at a similar level, after which it increases again after a longer or shorter time. The rapid increase in carbon monoxide concentration in the first phase of fire is consistent with the results obtained from previous combustion tests of a sample of the same pine wood in a cone calorimeter. However, due to different process scales, it occurs faster during a full- than small-scale fire. A temporary decrease in the carbon monoxide concentration is primarily due to the rapid reduction in

heat release rate (HRR) after reaching the maximum value and stabilization of the mass loss value (MLR). The lesser effect on that could be the flow of air stream in the ceiling zone resulting from the difference in pressure and convective movements arising during the combustion. The renewed increase in the CO concentration in the third phase (after approximately 200–300 seconds) resulted, on the one hand, from a further increase in the rate of heat release, characteristic of the wood burning process (release of more volatile products due to cracking of the carbonized layer), while, on the other hand, from incomplete combustion caused by insufficient oxygen in a closed room. From the graphs presented, the moment of extinguishing began in the second phase during the decrease in carbon monoxide concentration and it cannot be conclusively determined whether the extinguishing had an effect on the change in this concentration, especially that after approximately 300 seconds (in the case of air a little earlier), the CO concentration steadily increases, reaching after approximately 400 seconds the value of 500 ppm. The data in Table 3 show that virtually all selected tests have a concentration of 100 ppm already exceeded in the second minute of the fire before starting the extinguishing. It was fastest during test No. 23 (57 seconds) and slowest during test No. 1 (117 seconds). A concentration equal to the critical value of 500 ppm was achieved in the time range from 416 to 500 seconds from the moment of ignition. It was fastest achieved during test No. 16 (water with flow rate of 1.5 dm$^3$/min driven by air) and the slowest during test No. 20 (extinguishing using pure nitrogen without water). In this case, it is difficult to clearly determine the effect of extinguishing on these parameters. This is indicated by, among others, the fact that for water flow rate of 3 dm$^3$/min, the same time $\tau_{CO\,500}$ equal to 448 seconds, was obtained when using nitrogen or air.

Considering the graphs shown in Figures 9 and 10 and the data contained in Table 3, it follows that the oxygen concentration during extinguishing tests can be divided into two phases: in the first, lasting up to approximately 150 seconds or slightly longer, it is gradually reduced to approximately 16%; in the second, it remains at a more or less stable level. Oxygen concentration levels below the critical value of 15% (minimum equal to 12.8%) were obtained only for extinguishing when nitrogen without water mist was used. In other cases, this concentration was higher during the whole test. When extinguishing using air as a propellant gas (additional room aeration), the oxygen concentration began to decrease approximately linearly almost after ignition to stabilize at 16% after approximately 450 seconds. The change in water flow rate had practically no effect on the oxygen concentration. Its average rate of decline of approximately 0.011%/s for both water mist flow rates was almost twice lower compared to extinguishing with water mist driven by nitrogen. During the extinguishing only with nitrogen, the highest value of oxygen concentration rate, 0.024%/s, was obtained. In this case, the time to reduce the oxygen concentration to the critical value was equal to 476 s.

Errors in the estimation of measured concentrations of carbon monoxide and oxygen result from the accuracy of the sensors themselves, delays in their measurements as well as physical processes occurring in their immediate vicinity (e.g., air flows and high temperature).

## 5. Conclusions

Based on the analysis of the results included in Section 4, the following general conclusions have been formulated:

1.  The maximum extinguishing time for any tested hybrid system configuration did not exceed 8 minutes, which is in accordance with the requirements of FM Global 5580 approval.
2.  Considering quantities such as extinguishing time and temperature drop rate when assessing the extinguishing efficiency of the analyzed hybrid system, it can be concluded that the best configuration in this respect was the extinguishing system in which nitrogen-driven water fog was fed with a flow rate of 3 dm$^3$/min (the shortest extinguishing time and the fastest temperature drop).
3.  One of the least effective was the hybrid fire-extinguishing system, in which pure nitrogen was used for the extinguishing (relatively long extinguishing time with the slowest temperature drop).

During this test, almost three times more nitrogen was used than in the case of extinguishing with water fog.

4. The tests showed that the propellant gas used has an effect on the extinguishing process, which is not in line with the NFPA 750 standard.

5. Based on the obtained test results, it is impossible to determine any correlation between the extinguishing system configuration (gas type, water mist flow rate) and times of exceeding the two critical levels of carbon monoxide. The concentration obtained is more dependent on the fire development than the extinguishing process itself.

6. The carbon monoxide concentration already in the second minute from the moment of ignition exceeds the first critical level, although symptoms of poisoning at this concentration may appear only after prolonged stay in such an environment. Exceeding the second higher critical level occurs after approximately 7 minutes from the moment of ignition, which seems to be sufficient time to leave the endangered room.

7. The reduction in oxygen concentration in a room to a level that may have a negative impact on people occurs only after approximately 6 minutes of the fire (below 17%), which should be sufficient time to evacuate the endangered room.

8. Taking into account the complexity of the phenomena studied, based on the standard deviation of extinguishing times calculated for three different parameters of the hybrid extinguishing system (in the range from 17 to 22 s), it can be stated that the accuracy of its determination is relatively high and amounts to approximately 10%. The small range in which the standard deviation is included suggests that also for the other two cases, for which the number of tests was necessarily too small, the accuracy of the estimation of extinguishing time, as the basic parameter determining the extinguishing efficiency, will be comparable.

**Author Contributions:** Conceptualization, J.G. and P.W.; methodology, J.G. and P.W.; introduction and literature review, T.D.; formal analysis, J.G.; investigation, J.G. and P.W.; resources, T.D.; discussion, J.G.; conclusions, J.G. and T.D., data curation, J.G.; writing—original draft preparation, J.G. and T.D.; writing—review and editing, J.G.; supervision, J.G. and T.D.; funding acquisition, P.W.

**Funding:** Statutory project financed from means of MNiSW for the Main School of Fire Service No. S/E-422/31/17/18, entitled "Analysis of Extinguishing Effectiveness of Sprayed Jets Generated by Selected Mist Nozzles, Fog Nozzles Type Turbo and Extinguishing Lances During the Extinguishing of Type A Fires in a Closed Premise". Stage I, Warsaw 2017–2018, Project Manager T.D. (co-author of report T.D and J.G.).

**Conflicts of Interest:** The authors declare no conflict of interest. The funders had no role in the design of the study; in the collection, analyses, or interpretation of data; in the writing of the manuscript, or in the decision to publish the results.

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
