# Peer review of "Analysis of the Impact of Selected Parameters of the Hybrid Extinguishing System on the Fire Environment in a Closed Room"

_sustainability, doi:10.3390/su11236867_

Round 1

Reviewer 1 Report

This article presents tests regarding hybrid extinguishing system using water and air/or/N2. Several tests are presented and discussed. Results demonstrate the effect of N2 and water flow rate on the extinguishing time, CO production and O2 reductions.

The paper is clear and well-written. Objectives, context, methodology and results are clearly present and the conclusions are supported by the analysis.

The results are interesting. However, deeper analysis or clarification can be expected regarding the effect on temperature, CO and O2 concentrations.

Comments and remarks:

Line 124 : “[26].”, is the dot necessary ? Table 1 : could you indicate the k factor of the nozzle with water (q/(dp)^0.5) Line 149 : height of the compartment? Line 170 : indicate on a figure, the positions of the thermocouples Could you indicate the errors of thermocouples? Could you some comments about the fact that thermocouple’s end should be in contact with water drops and thus measured water temperature instead of gas temperature? Line 205 : how the approximation of 100kW is obtained ? give more details Line 211 : Could you give an estimate of the room tightness with the door closed? Line 226: Could you explain why there are 37 tests and only 5 are considered? Line 226 : In the results section, is it possible to have an example of results showing on the same figure the 36 thermocouple’s signal (first axis) and the CO and O2 Concentrations (on a secondary axis). It could help the reader to better understand the experiment. Line 308 : figures 6 and 7 instead of 5 and 6 Line 308-316: Could you define what you call “the average rate of temperature drop”? How it is computed? Line 308-316 : the analysis is unclear regarding the effect on gas temperature Figure 8 and 9 : CO concentration is given in kg/kg. In the text, it is discussed in term of ppm (mol/mol or vol/vol) which I guess is different as kg/kg. Could use the same unit? Lines317-336 : Could make clear that the first phase of CO increase corresponds to the combustion phase before the begin of the extinguishing system? Figure 8 and 9 : Could indicate on the graph the instant the extinguishing system starts? Lines317-336 : the effect of CO is unclear; Could you improve this section? What is the reason of the decrease and the increase of CO during the extinguishing phase? I guess that the decrease is due to the reduction of the fire HRR (or MLR) and the increase is due to the bad combustion. Figure 9-10 : Are you sure about the unit (kg/kg) ? And not %vol (mol/mol*100)? Figure 9-10 : Could you enlarge the y axis range? (12-22)% for instance? In order to clearly see the two stages : reduction of O2 and stationary level. Figure 9-10 : as for CO, indicate on the graph, the time the extinguishing system starts? Could you give the uncertainty of O2 and CO measurements? In order to correlate the effect of CO and O2, it could be interesting to plot on the same graph the two species.

Author Response

Dear reviewer,

Thank you very much for the positive and detailed review. I am grateful for your comments and suggestions. Below you can find answers for every comment.

Line 124: The error in the statement has been corrected.

Line 144: Flow factor was included in the table 1.

Line 149: Height of the compartment was given in the separate statement.

Line 170: The positions of the thermocouples have been added. They are shown in Fig. 3 and 4. The work analyzed the temperature measured by thermocouple No. 17, which was mounted under the ceiling above the extinguishing nozzles, from which the stream was directed downwards. From this it follows that this thermocouple was not in its path and so certainly the temperature of the air and not the water was measured.

Line 205: Explanation on fire power estimation has been added on lines 217-218.

Line 226: Lines 245-248 and 252-257 provide an explanation of why only 5 out of 37 tests were selected. In my opinion, the combination of temperature and CO and O2 graphs will significantly reduce their readability, because in this case the drawing will contain a large number of curves located close to each other. In addition, in this case, three and not two axes would have to be used (three different scales deg C, ppm and kg / kg). However, in the paper instead of four drawings (6a, 6b, 7a, 7b) containing temperature history, only one is placed, on which all interesting curves are marked with colored lines.

Line 308-316: The average rate of temperature drop is defined in Line 273-277 (expression No. 1).

In the version of the paper submitted for review, an error occurred in the drawings showing CO concentrations. The unit of concentration for CO on the vertical axis should be ppm and not kg / kg. In such units, this quantity was measured using the sensors used. The error in the graphs containing CO concentrations has been corrected.

Line 317-336: CO increase in the first phase was explained more detailed in the Lines 349-354. The beginning of extinguishing was marked on the CO and O2 charts using vertical lines whose colors correspond to the appropriate curves. CO decrease in the second phase and increase in the third phase was explained in the Lines 356-360. Y-axis of O2 was enlarged. Its range is currently 12-22% instead of 0-25%. Errors in the estimation of measured concentrations are briefly explained in the Lines 388-390 (sensors accuracy was given earlier with its description in the section 2 – Lines 199-202). The number of the figures containing CO and O2 concentrations have been reduced from eight (Fig. 8a, 8b, 9a, 9b, 10a, 10b, 11a, 11b) to two (Fig. 7 and 8), so both CO and O2 concentrations are placed in the same drawing.

I hope the above explanation is enough for you.

Best regards

Reviewer 2 Report

This is a well-written paper that studies the hybrid extinguishing system and its effects on the fire behaviors in an enclosed room. The experimental study was well-designed and implemented and temperature and species concentration were recorded. It is very useful to understand and quantify the hybrid parameters when utilizing a hybrid extinguishing system.

It would be very useful to include all experimental data in table 2, in which only 5 tests were presented. I suggest the author add a compressive statistical study over the effects of the gas and water flow rate individually. It will help to develop a model to control and predict the extinguishing time in such room fire.

Author Response

Dear reviewer,

Thank you very much for the positive review. I am grateful for your comments and suggestions. Below you can find answer.

Considering them, I included the values of all significant input and output quantities in one table No. 2. In the improved work I explained in more detail the reason for taking only 5 studies for analysis. It results, among others, from the difficulties in carrying out statistical processing due to the complexity of the phenomena studied and the associated difficulty in obtaining adequate repeatability, as well as too few tests of the air-fed system (too little gas available). Due to the above, I limited the statistical analysis to providing in Table 3 average values and standard deviations for the extinguishing time as the basic parameter determining the extinguishing efficiency of the tested hybrid system.

I hope the above explanation is enough for you.

Best regards

Round 2

Reviewer 1 Report

The authors improve significantly the article an answer satisfactorily the comment.